# Electrochemical Immunosensors with PQQ-Decorated Carbon Nanotubes as Signal Labels for Electrocatalytic Oxidation of Tris(2-carboxyethyl)phosphine

**DOI:** 10.3390/nano11071757

**Published:** 2021-07-05

**Authors:** Xiaohua Ma, Dehua Deng, Ning Xia, Yuanqiang Hao, Lin Liu

**Affiliations:** 1Henan Key Laboratory of Biomolecular Recognition and Sensing, Shangqiu Normal University, Shangqiu 476000, China; maxiaohua@sqnu.edu.cn; 2College of Chemistry and Chemical Engineering, Anyang Normal University, Anyang 455000, China; ddh@aynu.edu.cn (D.D.); xianing82414@csu.edu.cn (N.X.)

**Keywords:** electrochemical immunosensor, nanocatalyst, redox cycling, quinone

## Abstract

Nanocatalysts are a promising alternative to natural enzymes as the signal labels of electrochemical biosensors. However, the surface modification of nanocatalysts and sensor electrodes with recognition elements and blockers may form a barrier to direct electron transfer, thus limiting the application of nanocatalysts in electrochemical immunoassays. Electron mediators can accelerate the electron transfer between nanocatalysts and electrodes. Nevertheless, it is hard to simultaneously achieve fast electron exchange between nanocatalysts and redox mediators as well as substrates. This work presents a scheme for the design of electrochemical immunosensors with nanocatalysts as signal labels, in which pyrroloquinoline quinone (PQQ) is the redox-active center of the nanocatalyst. PQQ was decorated on the surface of carbon nanotubes to catalyze the electrochemical oxidation of tris(2-carboxyethyl)phosphine (TCEP) with ferrocenylmethanol (FcM) as the electron mediator. With prostate-specific antigen (PSA) as the model analyte, the detection limit of the sandwich-type immunosensor was found to be 5 pg/mL. The keys to success for this scheme are the slow chemical reaction between TCEP and ferricinum ions, and the high turnover frequency between ferricinum ions, PQQ. and TCEP. This work should be valuable for designing of novel nanolabels and nanocatalytic schemes for electrochemical biosensors.

## 1. Introduction

Electrochemical immunosensors have been attractive for a broad range of applications because of their exceptional attributes, including high sensitivity and selectivity, rapid response, low cost, and compatibility with miniaturization [1,2,3,4]. As regards the immunosensing formats, sandwich-type structures are among the most popular schemes, especially for ultralow concentrations of analyte [5,6]. In this assay, the crucial step is to enhance the detection sensitivity by a signal amplification strategy. Traditionally, enzyme-based signal amplification by horseradish peroxidase (HRP) or glucose oxidase (GOx) is the most common approach to improve the sensitivity of electrochemical immunoassays. The natural enzymes allow for fast and selective catalytic reactions; however, their practical applications in electrochemical immunoassays are still limited due to their costly and complicated preparation process and relatively poor stability [1,3,7]. Another drawback of enzyme-based, sandwich-type immunoassays is that the active center embedded in the insulating peptide backbone of the enzyme label cannot approach the electrode sufficiently closely, thus limiting the direct electron exchange with the electrode. For this consideration, nanomaterials as signal labels without enzyme modification have numerous remarkable merits in terms of improving sensitivity [1,3,8,9]. 

In general, nanomaterials can be exploited as nanocarriers to load signal markers, or can be directly used as signal reporters. Recently, metallic nanomaterials that can accelerate the electron transfer rate and exhibit fascinating catalytic activity have been employed as electrocatalysts to enlarge the electrochemical signal and avoid the shortcomings of enzymes—such as thermal and environmental instability [10,11,12,13,14,15,16]. However, there are still at least two drawbacks with nanoelectrocatalysts as signal labels for fabricating sandwich-type immunosensors: (1) long distances (a few tens of nanometers) between the nanoelectrocatalysts and the electrode may limit the direct electron transfer, and (2) surface modification of nanoelectrocatalysts and sensor electrodes with recognition elements and blockers may depress the electrocatalytic activity and make the direct electron transfer less feasible [1,3,17]. Therefore, there still remains significant room to develop novel nanolabels or nanocatalytic schemes for constructing sandwich-type immunosensors with high simplicity and sensitivity.

To accelerate the electron transfer between redox enzyme labels and electrodes, an electron mediator is usually used to shuttle the electrons. However, for the signal labels of nanocatalysts (in particular metal nanocatalysts), the electrochemical immunoassay promoted by the electron-transfer mediator is rarely reported, since it is hard to simultaneously achieve fast electron change between nanocatalysts and redox mediators as well as substrates [18,19]. To obtain excellent analytical performances, three requirements must be met for the mediated electron transfer: (1) the electron mediator should exhibit both a fast electrochemical reaction at the sensing electrode and a fast catalytic reaction with the electrocatalysts; (2) the substrate should undergo a very slow electrochemical reaction at the electrode; and (3) the reaction between the electron mediator and the substrate should be very slow. Previously, several groups have investigated the electron-transfer mediators of “outer sphere to inner sphere” electrochemical–chemical–chemical (ECC) redox recycling, and found that the optimal outer-sphere species (mediator) and inner-sphere species (substrate) are dependent on the working electrode [20,21,22,23,24,25,26,27]. We investigated the ECC redox recycling at a self-assembled monolayer (SAM)-covered gold electrode, and examined the effect of various electron mediators and substrates [26]. It was found that ferrocene derivatives exhibit a reversible electrochemical reaction on the SAM-covered electrode, and ferricinum ions (the oxidation state of ferrocene) show a fast chemical reaction with hydroquinone, aminophenol, and cysteamine. Further investigations demonstrated that TCEP is electrochemically inert at the SAM-covered electrode, and shows relatively slow electron change with ferricinum ions.

Although metallic nanocatalysts have been widely exploited as alternative signal labels to native enzymes to amplify electrochemical signals, there are rarely reports on the use of nonmetallic nanocatalysts as signal labels. Pyrroloquinoline quinone (PQQ) is a redox cofactor adopting an addition–elimination catalytic mechanism in many PQQ-dependent enzymes. Recently, PQQ-decorated nanomaterials and electrodes have been exploited as biomimetic heterogeneous electrocatalysts [28,29,30,31,32,33]. The PQQ tags exposed on the nanomaterials’ surface are able to transfer electrons directly to the electrode’s surface. For instance, Long et al. suggested that an electrode modified with PQQ-decorated carbon nanotubes (denoted as PQQ–CNTs) exhibits high electrocatalytic activity toward tris(2-carboxyethyl)phosphine (TCEP) oxidation, with a turnover frequency (TOF) of 133 s^−1^ [29]. Inspired by this study, herein, we developed a sandwich-type immunosensor with PQQ–CNTs as the electrocatalytic label. The electron transfer of PQQ on CNTs was promoted by the redox mediator of ferrocenylmethanol (FcM). The analytical performances and applications of the proposed electrochemical immunosensors were investigated. This work may pave the way for a more rational design of nanolabels for electrochemical immunoassays.

## 2. Materials and Methods

### 2.1. Reagents and Apparatus

Prostate-specific antigen (PSA) and antibodies were provided by Linc-Bio Science Co. Ltd (Shanghai, China). PSA ELISA kits, bovine serum albumin (BSA), human serum albumin (HSA), IgG, alpha-fetoprotein (AFP), thrombin, TCEP, l-ethyl-3(3-(dimethylamino)propyl)carbodiimide (EDC), N-hydroxysulfosuccinimide (NHS), 11-mercapto-1-undecanol (MU), and 11-mercaptoundecanoic acid (MUA) were purchased from Sigma-Aldrich (Shanghai, China). Amino-functionalized single-walled carbon nanotubes (CNT-NH_2_, 0.61wt % NH_2_-dopping) were obtained from XFNANO Materials Tech Co., Ltd. (Nanjing, China). Pyrroloquinoline quinine (PQQ), ferrocenylmethanol (FcM), and other reagents were supplied by Aladdin Industrial Corporation (Shanghai, China). Male serum was provided by the health center of Anyang Normal University (Anyang, China). All aqueous solutions were prepared with ultrapure water treated using a Milli-Q ultrapure water system.

Electrochemical measurements were carried out on a CHI660E electrochemical workstation with a three-electrode system comprised of a gold working electrode, a Pt counter electrode, and a Ag/AgCl reference electrode. The morphology of functionalized CNTs was characterized using a Hitachi SU8010 scanning electron microscope (Tokyo, Japan) and an FEI Tecnai G2 T20 transmission electron microscope (Hillsboro, OR, USA).

### 2.2. Preparation of PQQ–CNT-Ab_2_

CNT-NH_2_ was modified with antibody and PQQ through an amino covalent coupling reaction [29]. Prior to the modification, CNT-NH_2_ was dispersed in 4 M HCl and heated at 60 °C for 12 h. The treated CNT-NH_2_ was centrifuged, washed with ultrapure water, and then dried under vacuum overnight. After that, the purified CNT-NH_2_ (1 mg) was dispersed in 1 mL of DMF by ultrasonication for 1 h. Then, 1 mL of DMF solution containing 5 mM PQQ and 15 mM EDC was added to the CNT-NH_2_ suspension for shaking overnight. The resulting PQQ–CNTs were purified by centrifugation at 14,000 rpm and washing with DMF and water. The immobilization number of PQQ on the surface of CNT-NH_2_ was calculated by measuring the adsorption intensity change of PQQ at 330 nm with a standard curve method (Appendix A) [34]. The immobilization capability of CNT-NH_2_ to PQQ was estimated to be 287 ± 26 nmol/mg. The integration of PQQ on the CNTs was also confirmed by Fourier transform infrared (FTIR) spectroscopy (Appendix A). The second antibody of Ab_2_ was modified on the PQQ–CNTs’ surface through the EDC/NHS-mediated amino covalent coupling reaction between Ab_2_ and PQQ on the CNTs’ surface. In brief, the obtained PQQ–CNTs were dispersed in phosphate buffer solution (10 mM, pH 7.4) at a concentration of 1 mg/mL. Then, 0.5 mL of the PQQ–CNT suspension was mixed with 0.5 mL of phosphate buffer solution containing 0.2% Triton X, 0.4 M EDC, and 0.1 M NHS. After shaking for 15 min, the activated PQQ–CNTs were washed three times with the Triton-X-containing phosphate buffer solution. Then, 50 μL of 0.5 μg/mL Ab_2_ solution was added to 1 mL of the activated PQQ–CNT suspension. After shaking for 2 h at room temperature, the excess Ab_2_ was removed by three centrifugation/washing cycles at 12,000 rpm. The resulting PQQ–CNT-Ab_2_ was characterized by scanning electron microscopy (SEM) and transmission electron microscopy (TEM) (Appendix A). The obtained PQQ–CNT-Ab_2_ was dispersed in 1 mL of phosphate buffer solution containing 0.2% Triton X and 0.1% (*w*/*v*) BSA and stored at 4 °C for 12 h. Herein, BSA was used to block the unreacted sites and eliminate the nonspecific adsorption.

### 2.3. Preparation of Sensor Electrode

Gold disk electrodes were polished with alumina paste down to 0.05 μm, and then sonicated for 30 s in 50% ethanol. Capture antibody of Ab_1_ was immobilized on the surface of a gold electrode covered with MU/MUA mixed SAMs through the EDC/NHS-mediated amino covalent coupling. The SAMs were formed by incubating the cleaned gold electrode with an ethanol solution comprising optimized concentrations of 0.7 mM MU and 0.3 mM MUA (Appendix A). After washing with ethanol and water, the SAMs were activated by soaking the electrode in a solution comprising 0.4 M EDC and 0.1 M NHS for 15 min. After rinsing with water, the electrode was immersed in 0.1 mg/mL Ab_1_ solution for 12 h, and then incubated in a solution of 0.1 mM ethanolamine for 30 min to block the unreacted sites. To evaluate the stability, the antibody-modified electrode was stored at 4 °C in a clean environment.

### 2.4. Procedure for PSA Detection

10 μL of phosphate buffer solution containing a certain concentration of PSA (10 mM, pH 7.4) was cast on the Ab_1_-covered electrode at room temperature. After incubation for 30 min, the electrode was rinsed with ultrapure water. Then, 10 μL of the prepared PQQ–CNT-Ab_2_ suspension was cast on the electrode’s surface. After incubation for a further 30 min, the sensor electrode was washed with ultrapure water to remove non-specific PQQ–CNT-Ab_2_, and then placed in the phosphate buffer solution (pH 6) containing TCEP and FcM for electrochemical measurements.

For the assays of PSA in real samples, fresh male blood was collected, stocked for 2 h at room temperature, and then centrifuged at 2000 rpm for 5 min. After that, 100 μL of serum was taken out and diluted by 10 times with the buffer. For the electrochemical determination, 20 μL of the diluted sample was mixed with 10 μL of PSA standard sample at a given concentration. The other procedures followed those for the PSA standard sample analysis.

## 3. Results and Discussion

### 3.1. Detection Principle

Amperometric immunosensors mainly include sandwich-like and competitive formats. In general, sandwich-like immunosensors show high sensitivity and specificity because of their low background current and their use of a couple of specific antibodies [5]. Thus, the electrochemical immunoassays were carried out using the classical, sandwich-like format in this work (Scheme 1). PSA was tested as the model target. Carbon nanotubes were employed as the carrier of PQQ and Ab_2_ for signal readout. Insulating SAMs are widely utilized to immobilize capture elements—including antibodies—at the electrode–solution interface. The captured antibodies at the SAMs’ surface maintain their activity and specificity for target binding. Moreover, the SAMs with long linear molecules can eliminate the nonspecific adsorption, thus improving the specificity of affinity biosensors [35,36]. Herein, the mixed SAMs of MU/MUA with 11 alkyl carbons were used for the immobilization of capture antibodies, in which the MUA molecules allowed for the anchor of the antibodies via amino covalent coupling. The MU molecules regulated the surface density of MUA to prevent the formation of anhydride and N-acylurea between the neighboring carboxylic acids, and to facilitate the antibody–antigen interaction by weakening the steric hindrance effect [37,38]. In the detection step, FcM as the electron mediator can initiate the redox cycling between TCEP and PQQ–CNT, since the ferrocene derivatives undergo fast electron transfer at the SAM-covered electrode—even with 11 carbon spacers—and the oxidized ferrocene moieties (ferricinum ions) exhibit fast reaction with hydroquinone or PQQ, but show slow reaction with TCEP. In the so-called “outer sphere to inner sphere” ECC redox cycling system, PQQ–CNTs act as the biomimetic heterogeneous electrocatalyst for the oxidation of TCEP. Specifically, PQQ was first reduced to pyrroloquinoline quinol (PQQH_2_) by TCEP. Once FcM was electrochemically oxidized into FcM^+^ (the oxidation state of FcM), it would be regenerated by PQQH_2_, thus resulting in the increase in the anodic current.

### 3.2. Feasibility of the Method

Previous investigation has demonstrated that a carbon-fiber ultramicroelectrode modified with PQQ–CNTs exhibits high catalytic activity toward the electrochemical oxidation of TCEP through single-nanoparticle collision [29]. We first attempted to record the electrochemical signal by the direct electron transfer reaction between PQQ–CNTs and the electrode (Figure 1A). However, in the TCEP-free electrolyte solution, no clear redox wave was observed at the sensor electrode covered with PSA and PQQ–CNT-Ab_2_ (curve a). This indicates that the PQQ–CNT label exhibited poor electrochemical signal or electron transfer ability on the sensor electrode. We also monitored the change in the charge-transfer resistance of the sensor electrode before and after the capture of PSA (Appendix A). However, both cyclic voltammetry and electrochemical impedance spectroscopy (EIS) are insufficiently accurate and sensitive to monitor the change in charge-transfer resistance. These results can be ascribed to the fact that the recognition elements and blockers on the surface of the electrode as well as the CNTs, and/or the long distance between the PQQ and the electrode, limit the direct electron transfer of PQQ or [Fe(CN)_6_]^3^^−/4^^−^. Addition of TCEP to the electrolyte solution induced a clear enhancement in the anodic current within the potential range of 0.2–0.6 V (curve b). Note that the sensor electrode without capture of PQQ–CNT-Ab_2_ labels had no contribution to this reaction (curve c); thus, the enhancement of the anodic current in curve b can be attributed to the electrocatalytic oxidation of TCEP by PQQ–CNTs [29]. We also found that no significant signal enhancement was observed when the antibody–PQQ conjugate was used as the label instead of PQQ–CNT-Ab_2_ for signal output, demonstrating that the signal was amplified by using CNTs to load large quantities of PQQ.

To facilitate the electron transfer of PQQ and amplify the electrochemical signal, a well-known electron mediator of FcM was added. Figure 1B depicts the cyclic voltammograms (CVs) of the sensor electrode after the capture of PSA and PQQ–CNT-Ab_2_ in the redox-mediator-containing electrolyte solution. The Ab_1_-modified electrode, after the capture of PSA, only exhibited a couple of well-defined redox waves in the TCEP-free (curve a) and TCEP-containing (curve b) solutions of FcM, which are attributable to the oxidation and reduction of FcM. The reversible redox wave in the presence of TCEP suggested that the electro-oxidized FcM (FcM^+^) showed slow chemical reaction with TCEP. After the capture of PSA and PQQ–CNT-Ab_2_, the sensor electrode showed an irreversible electrocatalytic wave in the FcM solution (curve c). This indicates that FcM was regenerated by the PQQ–CNT label after its electro-oxidation. A much larger anodic current was obtained after the addition of TCEP to the FcM solution (curve d); this suggests that the electrochemical–chemical (EC) redox cycling between FcM and PQQ was promoted by the chemical–chemical (CC) redox cycling between PQQ and TCEP, thus amplifying the electrochemical signal.

### 3.3. Optimization of Experimental Conditions

The reaction rate between FcM^+^ and PQQH_2_ or PQQ and TCEP is constant during the electrochemical scan. However, the electrochemical reaction rate of FcM/FcM^+^ is dependent on the scan rate. Thus, a slow electrochemical scan can facilitate chemical redox cycling for FcM^+^/PQQH_2_ and PQQ/TCEP, thus improving the sensitivity of the “outer-sphere to inner sphere” ECC redox cycling system. Nevertheless, a slow electrochemical rate may allow for the occurrence of chemical reactions between the FcM^+^ and TCTP. For this consideration, we investigated the voltammetric characteristics of FcM at the SAM-covered electrode in the TCEP-free and TCEP-containing solutions (Appendix A). It was found that the presence of TCEP did not cause any significant change in the voltammetric characteristics of FcM/FcM^+^ at a scan rate of 20 mV/s or higher. However, TCEP induced a significant increase in the anodic current of FcM/FcM^+^ at a scan rate below 10 mV/s, which was accompanied by a decrease in the cathodic current. Thus, a scan rate of 20 mV/s was used in the ECC redox cycling system. 

Ferrocene derivatives as the general electron mediators for many oxidoreductases—including PQQ-dependent redox enzymes—can be modified on the electrode surface or dispersed in the solution to electronically “wire” the redox center upon contact and catalyze the oxidation of the substrate [39,40]. We also investigated the effect of FcM concentration on the peak current, and found that the current increased with the increase in FcM concentration. This suggests that high concentrations of electron mediators can facilitate the electron transfer reaction and the electrocatalytic oxidation of TCEP. However, high concentrations of electron mediators may also cause a high background current, thus decreasing the detection sensitivity. For this consideration, the effect of FcM concentration on analytical performance was evaluated by measuring the change in the redox-mediated catalytic anodic current (Δ*i*_pa_) at 0.4 V. The Δ*i*_pa_ increased greatly with the increase in FcM concentration until 50 μM (Appendix A), and then began to level off or even decrease slightly. Therefore, the electron mediator concentration was optimized to be 50 μM. Moreover, the effect of TCEP concentration on the Δ*i*_pa_ was also investigated. The Δ*i*_pa_ increased gradually with the increase in TCEP concentration, and began to level off beyond 200 μM (Appendix A). However, aside from the scan rate discussed above, we found that high concentrations of TCEP also showed an important influence on the voltammetric characteristics of FcM alone. At a scan rate of 20 mV/s, an increased anodic current accompanied by a decreased cathodic current was observed when TCEP concentration was higher than 250 μM (Appendix A). This demonstrates that high concentrations of TCEP can facilitate the chemical reaction between FcM^+^ and TCEP, thus leading to an increase in the background current and depressing the redox cycling between FcM^+^ and PQQH_2_. Here, a slightly excess concentration of TCEP (250 μM) was used. 

The density of MUA molecules assembled on the electrode surface may play an important role in both the activation of carboxylic acids for the immobilization of the capture antibody Ab_1_, and the followed-up capture of PSA and PQQ–CNT-Ab_2_ labels. In fact, we first investigated the effect of the MUA/MU ratio on the Δ*i*_pa_ (Appendix A), and found that the electrode modified with MUA/MU at a ratio of 3:7 exhibited the highest current change. Ab_1_ was immobilized on the electrode surface through the EDC/NHS-mediated amino covalent coupling. The concentration of Ab_1_ was optimized, as shown in Appendix A. To obtain a wide linear range, an excess concentration of Ab_1_ was used during the immobilization step. To confirm that the catalytic current was caused by PQQ assembled on the CNTs’ surface, the concentration of PQQ used for the preparation of PQQ–CNTs was optimized (Appendix A). To amplify the signal as much as possible, an excess concentration of PQQ was used for the preparation of PQQ–CNTs.

### 3.4. Sensitivity

Linear sweep voltammetry (LSV) is simple, sensitive, and rapid for quantitative analysis. To investigate the analytical merits of the biosensor, the LSV responses for the analysis of various concentrations of PSA were collected. As presented in Figure 2A, the currents were intensified with the increase in PSA concentration in the range of 0–5 ng/mL, demonstrating that the number of PQQ–CNT labels attached on the electrode surface is dependent on PSA concentration. Figure 2B depicts the calibration plot of the Δ*i*_pa_ at 0.4 V against PSA concentration. The sensor has a linear detection range of 0.005–1 ng/mL. The linear equation is expressed as Δ*i*_pa_ = 0.702(PSA) (ng/mL) + 0.014. The relative standard deviations (RSDs) obtained at three parallel prepared electrodes are all below 11.7% for the determination of various concentrations of PSA, suggesting an acceptable reproducibility. Moreover, the limit of detection (LOD) was estimated to be 5 pg/mL. The LOD is comparable to or even lower than that achieved by other immunosensors via the signal amplification of nanocatalyst labels (Appendix A) [41,42,43,44,45,46]. The high sensitivity can be attributed to the low background signal of the TCEP substrate, the signal amplification with CNTs as the carrier, and the high turnover frequency between FcM^+^ and PQQH_2_ as well as PQQ and TCEP. We believe that the analytical performances—such as detection sensitivity and linear range—may be improved by employing nanomaterial-modified electrodes for the immobilization of the capture antibody, and/or by using other PQQ-decorated nanocatalysts as the signal labels. Although the analytical performances were not good enough compared with those of the reported PSA electrochemical immunosensors, this work presents a novel nanolabel or nanocatalytic scheme for constructing sandwich-type immunosensor. For example, metal–organic frameworks (MOFs) containing a high density of metal centers have received tremendous attention in the fields of electronic and optical sensing. However, the poor electronic conductivity of MOFs and the long distance between the metal active centers and the electrode limit their application as the signal labels in electrochemical immunoassays, although some MOFs exhibit high biomimetic catalytic activity. Efforts are being made in our research group to synthesize PQQ-based MOF signal labels and develop electrochemical biosensors based on the “outer sphere to inner sphere” ECC redox cycling scheme.

### 3.5. Selectivity and Stability

In order to study the selectivity of the immunosensor, the LSV responses for the analysis of four common proteins (IgG, AFP, thrombin, and HSA) were collected. As a result, the proposed biosensor had no obvious response to the four proteins, even when their concentration was nearly 100 times higher than that of PSA (Figure 3). The result showed that the immunosensor exhibited high selectivity for PSA detection. Furthermore, the anti-interference ability was studied by incubating the sensor electrode with a PSA sample containing the four coexisting proteins. Consequently, no remarkable difference in the current change was observed in contrast to that in the presence of PSA only; this indicates that the immunosensor is highly specific for PSA analysis. The high selectivity and good anti-interference ability can be ascribed to the specific antibody–antigen interaction in the sandwich-like detection format, and the low non-specific adsorption of the PQQ–CNT-Ab_2_ label to the sensor electrode. Moreover, the stability of the sensor electrode is critical for the practical applications of electrochemical immunosensors. In this work, the stability of the immunosensor was investigated by monitoring the current response of the sensor electrode after storage at 4 °C for a given time. After one week, no significant current change was observed. After one month of storage, the sensor electrode lost only 6.7% of its capture efficiency. This good long-term stability can be attributed to the good stability and high activity of the antibodies attached to the electrode surface by the amino covalent coupling reaction. We also investigated the electrode’s regeneration, and found that the sensor electrode can be conveniently regenerated by immersing it in 10 mM NaOH solution (i.e., desorbing PSA and PQQ–CNT-Ab_2_). After eight regenerations, the current decreased by only 9.2% (Appendix A).

### 3.6. Serum Sample Assays

The concentration of PSA released into the circulatory system by a healthy prostate is less than 4 ng/mL. However, elevated levels of PSA have been found in the sera of prostate patients. To indicate the usefulness of our sensing strategy for clinical applications, human serum samples were tested. Figure 4 depicts the LSV responses of the sensor electrode for the analysis of diluted serum spiked with and without PSA standard samples. The current for the serum sample is higher than that for the buffer blank, indicating that the serum contains a certain amount of PSA. With the aforementioned established standard curve, the PSA concentration was calculated to be 0.048 ng/mL in the diluted serum or 0.72 ng/mL in the original serum; this suggests that the donor’s prostate was healthy. The PSA contents found in the PSA-spiked samples were close to the total values of the initiated and spiked PSA (Table 1). To evaluate the accuracy of our method, the PSA concentration was further quantified by a standard addition method. After spiking of three known concentrations of PSA to the diluted serum, the current was intensified with increasing concentration of PSA. Interestingly, the PSA content in the diluted serum sample spiked without PSA was found to be 0.073 ng/mL; this value is close to that achieved by the standard curve method, and indicates that the PSA concentration change in serum can be readily measured by the immunosensor. To check the correctness of the above results, PSA in the serum sample was also determined using a commercial ELISA kit. The values found by the proposed immunosensor and the ELISA kit were of the same order of magnitude; thus, the immunosensor is suitable for the detection of PSA in real samples, and may provide a useful means for clinical research. We noticed that the values obtained by this method were statistically lower than those determined by the ELISA kit. A possible explanation for this result is that the biological matrices may adsorb on the electrode surface to decrease the analytical performance.

## 4. Conclusions

This work presented a novel strategy for the development of non-enzymatic electrochemical immunosensors, in which nonmetallic PQQ-decorated CNTs were used as the biomimetic heterogeneous nanocatalyst for TCEP oxidation. The mixed SAMs with long linear molecules facilitated the immobilization of capture antibodies and eliminated the nonspecific adsorption. The electron transfer was promoted by the extra redox mediator of FcM. The feasibility and sensitivity of the proposed strategy were demonstrated with PSA as the model protein. The proposed immunosensor was also used to quantify PSA presented in the human serum sample. The results were consistent with those obtained by the commercial ELISA kit. Although the accuracy and sensitivity of the immunosensor must be improved, this work should be valuable for the design of novel nanocatalysts and electrochemical sensing platforms. Efforts are being made in our group to prepare new PQQ-based nanocatalysts with uniform size, and to develop novel sensing schemes using PQQ-mediated redox cycling.

## Data Availability

Not applicable.

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
