# Peer review of "Electrochemical Immunosensors with PQQ-Decorated Carbon Nanotubes as Signal Labels for Electrocatalytic Oxidation of Tris(2-carboxyethyl)phosphine"

_nanomaterials, 2021, doi:10.3390/nano11071757_

Round 1
Reviewer 1 Report
The manuscripts describes a new electrochemical assay for the detection of a prostate specific antigen (PSA) as the model analyte. The immunosensor architecture is rather complex , with the second antibody marked with PQQ-decorated CNT, as signal label for the electrocatalytic oxidation of 3 Tris(2-carboxyethyl)phosphine in the presence of a second mediator ferrocenylmethanol (FcM). Despite the principle of detection being complex, the study is well conducted, with the influence of several parameters on the immunosensor response investigated and discussed. The immunosensor was also used to quantify PSA in the human serum sample and results compared with those obtained using a commercial ELISA kit.
I recommend the publication of the manuscript after some improvements:
1) demonstrate the formation of SAM and prove the importance of its presence to improve immunosensor selectivity - experiments in the presence and absence of SAM
2) demonstrate the immobilization of the first antibody - evaluate the amount of antibody immobilized/available for the biorecognition reaction
3) prove the importance in having CNT in the structure of the second antibody (experiments in the absence of CNT).
Author Response
We thank the reviewer for his/her positive comments: “The manuscripts describes a new electrochemical assay for the detection of a prostate specific antigen (PSA) as the model analyte. The immunosensor architecture is rather complex, with the second antibody marked with PQQ-decorated CNT, as signal label for the electrocatalytic oxidation of 3 Tris(2-carboxyethyl)phosphine in the presence of a second mediator ferrocenylmethanol (FcM). Despite the principle of detection being complex, the study is well conducted, with the influence of several parameters on the immunosensor response investigated and discussed. The immunosensor was also used to quantify PSA in the human serum sample and results compared with those obtained using a commercial ELISA kit. I recommend the publication of the manuscript after some improvements:”
Comment 1: “demonstrate the formation of SAM and prove the importance of its presence to improve immunosensor selectivity - experiments in the presence and absence of SAM.”
Response: The SAM was used to immobilize the capture antibody through the EDC/NHS-mediated amino covalent coupling, which is the general method for the immobilization of antibodies on the gold electrode. The capture antibody at the SAM surface maintain its activity and specificity for target binding. Moreover, the SAM can reduce or eliminate the nonspecific adsorption of biomolecules and decrease the background signal of TCEP in the “outer-sphere to inner-sphere” electrochemical-chemical-chemical (ECC) redox recycling. The importance of SAM for the design of immunosensor has been well investigated in our and other studies (Anal. Chem. 2008, 80, 2556; Colloids Surf. B: Biointerf. 1999, 15, 3; Electrochim. Acta 2014, 139, 323). We have cited the references and discussed the importance of SAM in the main text.
Comment 2: “demonstrate the immobilization of the first antibody - evaluate the amount of antibody immobilized/available for the biorecognition reaction.”
Response: It is a good question. The concentration of capture antibody has been optimized and the result has been shown in Supplementary Material (Fig. S10). To obtain a wide linear range, excess concentration of Ab1 was used during the immobilization step.
Comment 3: “prove the importance in having CNT in the structure of the second antibody (experiments in the absence of CNT).”
Response: It is an excellent comment. We first attempted to label the second antibody with PQQ and used the antibody-PQQ conjugate as the label for signal output. However, no signal enhancement was observed. Thus, carbon nanotube (CNT) was used to load PQQ molecules and to immobilize the second antibody.
Reviewer 2 Report
The authors described a new electrochemical immunosensor using carbon nanotubes functionalized with PQQ in order to detect PSA. Ability of the fabricated sensor was well evaluated by electrochemical measurements. The manuscript includes new and valuable information for researchers in immunosensors. I think the manuscript is acceptable after minor revision.
- S3. It looks that SEM features of CNT were changed after decoration with PQQ. Are there any changes in CNT physical and chemical properties?
- Stability of the sensor (line 346). The authors suggested 8 times regeneration was OK, but no data were indicated. Could you include the data of the regeneration?
- Please discuss about speed of the sensing.
Author Response
We thank the reviewer for his/her positive comments: “The authors described a new electrochemical immunosensor using carbon nanotubes functionalized with PQQ in order to detect PSA. Ability of the fabricated sensor was well evaluated by electrochemical measurements. The manuscript includes new and valuable information for researchers in immunosensors. I think the manuscript is acceptable after minor revision.”
Comment 1: “It looks that SEM features of CNT were changed after decoration with PQQ. Are there any changes in CNT physical and chemical properties?”
Response: No essential morphology change was obserbed by SEM before and after the modification of PQQ. The immobilization number of PQQ on the surface of CNT was calculated by measuring the adsorption intensity change of PQQ. The immobilization capability of CNT-NH2 to PQQ was estimated to be 287 ± 26 nmol/mg. The integration of PQQ on CNT was also confirmed by Fourier transform infrared (FTIR) spectrum. The results have been shown in Supplementary Material (Fig. S1 and S2).
Comment 2: “Stability of the sensor (line 346). The authors suggested 8 times regeneration was OK, but no data were indicated. Could you include the data of the regeneration?”
Response: We have discussed the result in the main text and added the data in Supplementary Material (Fig. S12).
Comment 3: “Please discuss about speed of the sensing.”
Response: Electrochemical biosensor shows the advantage of fast response. The experimental conditions for target detection have been shown in Part 2.4. Both the merits and defects of the biosensor have been discussed in Conclusion.
Round 2
Reviewer 1 Report
I am pleased with the alterations made.